# A Novel Autoencoder Based Approach for Counterfactual Estimation Using Sparsity Constraints

## Abstract

Building upon the abduction-action-step scheme and the structural causal model framework, this paper introduces the Conditional Sparse Autoencoder (CSAE), a novel approach for time series counterfactual estimation using encoder-decoder based architectures with a sparsity constraint to disentangle the roles of the inputs in the expected outputs. We benchmark CSAE with Conditional Variational Autoencoder (CVAE), the most widely adopted encoder-decoder architecture for counterfactual estimation, showing that CSAE clearly outperforms CVAE in this domain. Furthermore, we demonstrate the versatility of CSAE by extending it to image-based counterfactual scenarios, obtaining promising results. This work has important implications for a wide range of applications across various domains including finance, healthcare, and transportation, where being able to perform accurate counterfactual estimations is critical for decision-making.

## 1 Introduction

Understanding the complexities of causal reasoning is crucial for making sense of our world. This type of reasoning allows to analyse interactions with our environment (interventions) and hypothetical alternate worlds (counterfactuals). While fields like econometrics have long embraced causal inference methods (Wold, 1954), the inclusion of these techniques in the domain of deep learning (DL) is a more recent development (Kaddour et al., 2022; Schölkopf, 2022). Existing research in DL has primarily focused on pattern recognition and predictive modeling, often failing to distinguish between correlation and causation. This has resulted in DL models that are prone to biases (Zhao et al., 2017), vulnerable to changes in input distribution (Szegedy et al., 2014), and generally lacking in transparency (Kusner et al., 2017). Previous work has begun to address these gaps by integrating causal methods for tasks like causal disentanglement (Yang et al., 2022; Parascandolo et al., 2018), causal discovery (Sanchez et al., 2023; Goudet et al., 2018; Bengio et al., 2020), data augmentations (Gowda et al., 2021; Kaushik et al., 2020) or causality-based explanations (Parafita & Vitrià, 2019; Singla et al., 2020; Wu et al., 2023). More recently, some works have developed methods that allow to answer counterfactual questions (Kaddour et al., 2022).

Counterfactuals allow to reason about alternative realities by asking "what if" questions, and to quantify the effects of hypothetical interventions on an outcome of interest, as if we could "re-run" the world under different conditions. Works like Pawlowski et al. (2020) leverage deep learning prowess to estimate counterfactuals on the basis of Structural Causal Models (SCMs) and the abduction-intervention-prediction process (Pearl, 2000). These papers work with images and respond questions like "How would that brain scan be if the individual was 10 years older?", or "How would that face look if they were smiling?" (Pawlowski et al., 2020). However, these works often do not disentangle enough the effects of the causal attributes over the output and lack precision. For this reason, although counterfactual images are usually reasonable, when applying these methods to some time series settings where more precision is required, the results are usually not satisfactory.

Time Series counterfactuals, which are the main focus of this paper, can help in identifying the causal drivers of observed phenomena and in predicting the outcomes of interventions in many fields. For example, in finance (Barocas et al., 2020), counterfactual analysis can be used to estimate the effect of a hypothetical intervention, such as a change in interest rates, or in stock prices. In healthcare

(Prosperi et al., 2020), causal inference can help in identifying the causal factors of diseases and in evaluating the effectiveness of treatments (Zou et al., 2020) or, as the original industrial motivation of the present article, in cases where a competitor product is introduced into the market, it is essential to evaluate the counterfactual effect of the new product on the sales of the existing product, which could help determine the optimal pricing strategy and marketing tactics for the existing product.

In this context, our work adapts the framework of SCMs and abduction-action-prediction process to time series data, and introduces the Conditional Sparse Autoencoder (CSAE), a novel method for counterfactual inference which is designed specifically to disentangle the effects of the causal variables over the outcome. It has been conceived to allow sound time series counterfactuals and, as shown in this paper, has proven to outperform Conditional Variational Autoencoder (CVAE), the most used autoencoder based model for counterfactual inference. Additionally, we demonstrate that CSAE is also effective for image counterfactuals.

The present paper is organized as follows: Sec. 2 provides a comprehensive overview of the extant literature on counterfactual inference. Sec. 3, explicates the utilized methodology, which includes an exposition of causality concepts, a definition of the methods and an explanation of the counterfactual inference process within the context of our problem. Sec. 4 presents the datasets, models, metrics, and results. Finally, the conclusions drawn from our work are expounded upon in Sec. 5.

## 2 RELATED WORK

Many works have appeared recently in the intersection among causality and machine learning. The ones which are more close to ours are those about counterfactual estimation using deep generative models, which are usually applied to images. Pawlowski et al. (2020) show how to jointly model all the functional assignments in an SCM using deep generative models, and apply Variational Autoencoders (VAEs) (Kingma & Welling, 2022) and normalizing flows (Kobyzev et al., 2020). For the same purpose, Dash et al. (2022) and Shen et al. (2021) use GANs (Goodfellow et al., 2014) and Jeanneret et al. (2022) use diffusion models (Ulhaq et al., 2022). Another approach based on Graph Neural Networks is proposed in Sanchez-Martin et al. (2021). Kim et al. (2020) proposes a VAE-based approach that clusters the causal graph based on which features undergo interventions, which is useful in the context of complex causal graphs with sensitive variables. Monteiro et al. (2023) presents some useful metrics to evaluate counterfactuals. Sauer & Geiger (2021) use deep neural networks to disentangle object shape, object texture and background in natural images. Van Looveren & Klaise (2021) utilize class prototypes in order to find interpretable counterfactual explanations. Parascandolo et al. (2017) use multiple competing models in order to retrieve a set of independent mechanisms from a set of transformed data points in an unsupervised way.

There are other interesting works about causal representation learning with deep generative models that do not tackle directly the problem of counterfactual inference but have a close relationship with this work. CausalGAN Kocaoglu et al. (2017) and Liu et al. (2019) combine GANs (Goodfellow et al., 2014) with SCMs, basing the generator architecture on an assumed causal graph. However, these methods lack tractable abduction capabilities and therefore cannot generate counterfactuals and reach only the second rung of the causal ladder (Pearl, 2000). Yang et al. (2022) propose a method that learns a causal model, including the DAG, over latent variables from data and generates counterfactual samples. Kumar et al. (2023) use a GAN based approach to address the specific problem of spurious correlations in medical datasets.

In the field of causal machine learning applied to time series, some methods have been proposed that either are not based on deep generative models or do not directly address the problem of counterfactual inference. Tonekaboni et al. (2020) proposes a method for time series explanation that they claim to be based on counterfactuals. However, their authors use forecast methods that do not take into account actual observations and perform something more similar to an intervention. Liu et al. (2022) imputes counterfactual outcomes for treated observations to estimate the average treatment effects (ATEs) using techniques like fixed effects counterfactual estimator, interactive fixed effects counterfactual estimator or matrix completion estimator. Bica et al. (2020) develops a method that leverages the assignment of multiple treatments over time to estimate treatment effects in the presence of multi-cause hidden confounders. Ahmad et al. (2021) propose a method to detect causal relationships in time series. ARCO (Carvalho et al., 2018) uses a combination of machine learning and time series econometrics techniques to estimate the causal effects of a treatment on a outcome

variable in a panel time series data setting when a single unit is treated and control group is not available. Brodersen et al. (2015) propose a method that estimates counterfactual time series in the presents of an event with two main information sources: the historical part of the time series, previous to the event, and other time series that were predictive of the target series prior to the event and have not been affected by it. In our view, the fact that these models rely on time series other than the target one represents a significant weakness, as it may not always be feasible to access data with a reasonable predictive capacity. For this reason, it is not very reasonable to compare these model with the ones that do not require external information sources, as the one we propose. On the other hand, we believe that the actual values over which the counterfactual is to be obtained bring an important information that should be used if it is possible, i.e. in the case a sufficient amount of event and event-less data for a model to learn to separate the effects of the event from the rest of effects in the actuals is available. For example, if something happens after and independently of the event that alters the observed values of a time series, the effects of that happening should be reflected in the counterfactual estimate.

Finally, there are other interesting counterfactual estimation approaches that are applied to tabular data. Among them, Yoon et al. (2018), based in GANs and Vlontzos et al. (2021), based in Deep Twin Networks, similar to siamese networks (Koch et al., 2015), stand out.

## 3 METHOD

This section begins with an overview of SCMs, followed by the introduction of CVAE, the most used autoencoder based model for counterfactuals, and the explanation of CSAE, our proposed model.

### 3.1 BACKGROUND ON STRUCTURAL CAUSAL MODELS

The proposed approach for counterfactual estimation is based on the definition of counterfactual provided by Pearl (2000), which corresponds to the third rung of the causation ladder. Counterfactuals can be operationalized by employing SCMs.

A SCM $\mathcal{M} := (\mathbf{S}, p(\epsilon))$ consists of a collection $\boldsymbol{S} = \{f_i\}_{i=1}^N$ of structural assignments $h_i := f_i(\epsilon_i; \mathbf{pa}_i)$, where $\mathbf{pa_i}$ is the set of parents of $h_i$ (its direct causes), and a joint distribution $p(\boldsymbol{\epsilon}) = \prod_{i=1}^N p(\epsilon_i)$ over mutually independent exogenous noise variables (i.e. unaccounted sources of variation) (Pawlowski et al., 2020). As the assignments are assumed acyclic, a directed acyclic graph (DAG) can represent relationships, with edges pointing from causes to effects in a causal graph. A unique joint observational distribution $P_{\mathcal{M}}(h)$ is determined by every SCM, fulfilling the causal Markov assumption: each variable is independent of its non-effects given its direct causes. Thus, it factorizes as $P_{\mathcal{M}}(h) = \prod_{i=1}^N (P_{\mathcal{M}}(h_i \mid \mathbf{pa}_i))$, where each conditional distribution $(h_i \mid \mathbf{pa}_i)$ is determined by its assignments and noise distribution (Peters et al., 2017).

SCMs allow to perform Counterfactual queries in a three-step procedure (Pearl, 2000): **1) Abduction**: predict the 'state of the world' (the exogenous noise $\epsilon$) that is compatible with the observed data $\mathbf{h}$, i.e. infer $P_{\mathcal{M}}(\boldsymbol{\epsilon} \mid \boldsymbol{h})$. Replacing the prior distribution of noise variables $p(\boldsymbol{\epsilon})$ by this posterior distribution, we obtain the counterfactual SCM $\mathcal{M}_h := (\mathbf{S}, p(\epsilon \mid h))$; **2) Action**: perform an intervention (i.e. $do(h_i := \widetilde{h}_i)$) to the counterfactual SCM which corresponds to the desired manipulation, which generates the modified counterfactual SCM $\widetilde{\mathcal{M}} := \mathcal{M}_{\mathbf{h}, do(\widetilde{h}_i)}$; **3) Prediction**: compute the quantity of interest based on the distribution entailed by the modified counterfactual SCM, $P_{\widetilde{\mathcal{M}}}(h)$. In the next section, we introduce autoencoder based approaches for counterfactual inference and explain how they approximate structural equations and abduct the exogenous noise.

### 3.2 AUTOENCODER BASED MODELS FOR COUNTERFACTUAL ESTIMATION

A deep learning model can be used to perform counterfactuals if at the same time it is expressive enough to approximate an structural equation and has abduction capabilities. Some autoencoder based architectures fulfill both conditions and therefore are suitable for counterfactual estimation. In this section, we first explain CVAE, the most used autoencoder based model for counterfactual inference, and then we describe CSAE, our proposed approach.

**CVAE:** The loss function for a classic CVAE (Kingma & Welling, 2022) with a $\beta$ penalty (Higgins et al., 2017), which correspond to the evidence lower bound (ELBO), is given by:

$$\mathcal{L}_{\text{CVAE}}(\theta, \phi) = \mathbb{E}_{q_\phi(z|x,\mathbf{pa})}[\log p_\theta(x|z, \mathbf{pa})] - \beta \, KL[q_\phi(z|x, \mathbf{pa}) \parallel p(z))] \tag{1}$$

where both $q_\phi(z|x, \mathbf{pa})$ and $p_\theta(x|z, \mathbf{pa})$ are a set of dimension-wise independent normal distributions parameterised, respectively, by an encoder neural network $E_\phi$ and a decoder neural network $D_\theta$, $p(z)$ is an isotropic normal prior distribution, KL is the Kullback–Leibler divergence (Hall, 1987) and $\beta$ is a penalization over KL (Higgins et al., 2017). In the model training, the ELBO function is maximized with respect to parameters of the neural networks using the re-parametrization trick to sample from the approximate latent posterior: $z = \mu_\phi(x, \mathbf{pa}) + \alpha_\phi(x, \mathbf{pa}) \odot \epsilon_z \sim \mathcal{N}(0, I)$.

It is possible to generate a counterfactual sample $x^*$ by encoding an observation $x$ and its parents $\mathbf{pa}$, i.e. obtaining the normal distribution $q_\phi(z|x, \mathbf{pa})$, where position and scale parameters come from the encoder: $\mu_\phi, \alpha_\phi = E_\phi(x, \mathbf{pa})$, then sampling the latent posterior from this distribution: $z \sim q_\phi(z|x, \mathbf{pa})$, and finally decoding it along with the counterfactual parents $\mathbf{pa}^*$: $x^* \sim p_\theta(x|z, \mathbf{pa}^*)$. Notice that, at a practical level, the counterfactual will be decoded from the latent sample in a deterministic way: $x^* = D_\theta(z, \mathbf{pa}^*)$.

There is a clear parallelism among this process and the abduction-action-prediction sequence previously defined, even if the latent variable $z$ is not the same as the exogenous noise $\epsilon$ of the SCM (Monteiro et al., 2023). Despite this, even if CVAE is a reasonable option to generate counterfactuals, there are some relevant problems. On the one hand, the model can ignore, completely or partially, the conditioning, which will produce that the latent variable $z$ accounts for factors of variation that correspond to the conditioners, and thus the counterfactuals will be poor. Additionally, this can make the decoder network not properly disentangle the effects of each parent on the observation. These disentanglement problems can be addressed by using a more narrow bottleneck (reducing the dimensionality of the latent space), by reducing the scale parameter of the prior distribution, or with a $\beta$ penalty on the KL. Nonetheless, all these measures solve only partially the described problems, introduce trade-offs among the counterfactual soundness and the reconstruction capabilities (which also affect the quality of the counterfactuals), and require an important effort of hyperparameter search. Next, we introduce a novel approach that aims to overcome these issues.

### 3.2.1 IMPROVING COUNTERFACTUAL ESTIMATION WITH CONDITIONAL SPARSE AUTOENCODER

A regular Autoencoder (AE) learns a low dimensional representation of a given input by jointly training an encoder $E_\phi$ that outputs a latent compressed representation of the input and a decoder $D_\theta$ that reconstructs the input from the latent variable. Even if it has the same encoder-decoder structure as a VAE, conditioning an AE would not be useful at all for the purpose of estimating counterfactuals, because if the latent space of the model is unconstrained, it will not use the conditionings at all and counterfactual estimates will be poor.

Unlike a regular AE, in a sparse autoencoder the network is trained to, together with the aforementioned objective of a regular AE, enforce sparsity in the learned representation, meaning that the values of this representation tend to be close to zero and only a small number of hidden units are activated for a given input. This is achieved by adding an L1 or L2 regularization term over the bottleneck latent variables to the AE loss. In this work, we propose to add a sparsity constraint to an conditional AE to infer counterfactuals. Although Sparse autoencoders are well known in the literature, to the best of our knowledge, conditioning them and using the sparse regularization to perform counterfactuals is a novel approach. This proposed model, which we call CSAE, has the following loss function:

$$\mathcal{L}_{\text{CSAE}}(\mathbf{x}) = |x - \hat{x}| - \lambda \sum_i |z_i| \tag{2}$$

where $x$ is the input variable, $z_i$ are the elements of the latent space vector $z = E_\phi(x, \mathbf{pa})$, $\lambda$ is the hyperparameter of the penalty term, which in this work we have chosen to be L1, and $\hat{x}$ is the reconstruction term which stems from the decoder: $\hat{x} = D_\theta(z, \mathbf{pa}^*)$. As in CSAE, there is a parallelism among the abduction-action-prediction scheme and how CSAE performs counterfactuals; thus, $z$ would make the function of the abducted exogenous noise and the action refers to the introduction of the counterfactual parents $\mathbf{pa}^*$ in the decoder instead of the factual parents $\mathbf{pa}$.

The quality of the estimated counterfactuals depends strongly on the capacity of the counterfactual model to disentangle the roles of the inputs in the expected output, a problem which is common in causal machine learning. Even if this is a relevant problem for all kinds of data, it is specially critic for some time series settings as the required precision of the counterfactual is specially high, thus making it key to effectively learn to separate the contributions of the parents from the rest of contributions of the data generating process. This allows to properly reflect in the counterfactual estimation any effects on the actual values of the time series that are independent of the parents.

To foster an AE based model's disentanglement ability, the capacity of the latent space to seize information must be limited in some way so that the latent varianle does not capture the information that the conditioning bring, which would distort counterfactual estimation. Thus, it is not enough that the decoder part of an autoencoder based model reconstructs properly the input given the latent variable and the parents (which condition the model), but it has to be able to efficiently separate the contributions of conditionings from the exogenous noise. In the case of CVAE, this is achieved partially by adjusting hyperparameters like the latent space dimensionality and the scale parameter of the prior distribution in order to limit the information that the latent space can capture while enabling an accurate reconstruction. As mentioned in the previous subsection, the problem with this approach is that there is a trade-off between disentanglement and reconstruction capacity and, furthermore, the probabilistic nature of CVAE introduces additional errors as the input of the decoder comes from a random sample of the posterior distribution that the encoder outputs and is not directly the outcome of the encoder.

CSAE has been conceived as a solution to the disentanglement problem that CVAE and most of the possible frameworks that could have been used feature. By introducing the sparsity loss in Equation 2, the latent variables are forced to be as close to zero as possible while the reconstruction capability of the model is almost unaltered. This implies that the latent space is forced to capture the minimum necessary information for the decoder to perform the reconstruction and, given that the parents condition the decoder, this implies to force the latent vectors to seize only the information that is not present in the parents, which amounts to forcing the model to use the conditionings.

## 4  EXPERIMENTAL SECTION

In this section, we explain the datasets where the models have been applied, the metrics that have been used to validate these models, the details of the models and baselines, and finally the results.

### 4.1  DATASETS FOR COUNTERFACTUAL ESTIMATION

#### 4.1.1  TIME SERIES COUNTERFACTUALS

In this work, we estimate time series counterfactuals in the presence of events. All the time series datasets share a common structure and can be described by the same simple causal graph. Let $h$ be the historical part of a time series previous to the entrance of an event $e$ which only affects the posterior steps of the time series, and $y$ the actual values over which we want to compute the counterfactual. $h$ and $e$ are the parents of $y$, and they are independent, i.e. there is no confounding. $e$ is a binary variable, where 0 indicates absence of event and 1 indicates presence of event. When computing the counterfactuals, $h$ remains constant and $e$ is the only intervened variable. We have applied our models to the following datasets:

**TS Synthetic dataset.**   We have created a synthetic dataset of time series with 30 steps that initiate always at time 0, have a trend that stems from a uniform distribution [-0.1,0.1] (meaning that this amount is added at each step), a drop of 0.7 in the step 20 in case of series with event, and an additional change at any randomly chosen step post-event with a value chosen uniformly within a range from -0.7 to 0.7. This value represents the effect of a happening that affects the time series both in case of event and without event, whereas the previous drop of 0.7 represents the effect of the main event. Besides, a gaussian noise $N(0, 0.1)$ is added to each step to make the time series more realistic. As can be noticed, the only difference in the generating process of these data for time series with and without event is the drop of 0.7, which allows to generate at the same time a time series with or without event and its counterfactual ground truth. Thus, it is possible to use any traditional metric to see how similar the counterfactual estimate is to the counterfactual ground truth.

**TS Semi-synthetic dataset.** This dataset is based in Rosseman Store Sales dataset rossmann, a public dataset from a Kaggle competition. It shows the daily sales of Rossmann drug stores from 2013 to 2015. We simulate a situation where the first Monday of every march from 2013 to 2015 includes an event that affects half of the stores (e.g. it could be a promotion, a marketing campaign, etc.) and multiplies the sales by 1.1 the first day, 1.2 the second day, and 1.3 the rest of the days during three weeks. We use the four weeks previous to the first Monday of march of each year as historical time series and the following three weeks as actual values. As in the case of the synthetic dataset, the simulated event allows to have a ground truth counterfactual. Rossman store sales are influenced by many factors, including promotions, competition, school and state holidays, seasonality, etc., and we want our counterfactual estimate to capture the effects of these factors in a better way than if we used just the historical information.

**TS Real world dataset.** This dataset shows the monthly sales of a pharma company and, as in the previous datasets, has two types of time series depending on whether an event has impacted them or not. The event here corresponds to the month where a generic treatment enters the market, which usually happens a few months after the date when the patent expires, which is not related with the features of the drug or its sales. Thus, to consider that there is no confounding among the event and other factors that alter sales is a good approximation to reality. We take 12 months as historical time steps and 30 months as actuals. This is a private dataset as it contains company specific information.

### 4.1.2 IMAGE COUNTERFACTUALS: COLOR-MNIST

We demonstrate the general applicability of our proposed model by performing image counterfactuals on color MNIST, a dataset based in MNIST dataset (LeCun & Cortes, 2010) that we construct following Monteiro et al. (2023).As in the paper, in addition to the digit we introduce a new parent: the digit's hue. We colour each image by triplicating the grey-scale channel, setting the saturation to 1 and the hue uniformly to a value between 0 and 1. The causal graph of this setting is simple: the digit and the hue are the parents of the image, and are independent, i.e. there is no confounding.

### 4.2 EVALUATED METRICS

In order to evaluate the performance of the proposed methods, we have used the following metrics, some of which, as explained in each paragraph, are particular for some datasets:

**MAE with the ground truth counterfactual.** We compute Mean Absolute Error (MAE) among the estimated counterfactual and the ground truth counterfactual for those settings where we know it. It is possible to know the counterfactual ground truth in those settings where we control the generative process over the parent of interest, i.e. synthetic and semi-synthetic time series datasets, and color-MNIST with respect to the hue parent.

**MBE with the ground truth counterfactual.** We calculate Mean Bias Error (MBE) among the estimated counterfactual and the ground truth counterfactual for the synthetic and semi-synthetic time series datasets, which allows to detect bias in the counterfactual estimations.

**Added Variations.** This metric has been conceived as a proxy to evaluate the reliability of a counterfactual estimate when the ground truth is not available, as it is the case of real world data. The core idea is that, regardless of the time series values of the counterfactual estimate, if the method is accurate, in the case that we introduce variations in the actuals, the model should understand that they are the effect of some process that has nothing to do with the event and, therefore, should reflect them in the counterfactual estimate. Thus, with this metric we can evaluate one property that such a method should feature: added variations in the actuals should be reflected in the counterfactual estimate. This metric is implemented as follows: for each actual to be evaluated, several positive and negative values in the order of the time series values are chosen; for each of this values, several windows of few consecutive steps from the actuals are selected and the chosen value is added to those steps. After that, a counterfactual estimate is obtained for every altered actual and it is compared to the counterfactual estimate of the non-altered actual. Two quantities are obtained:

- **Total difference:** takes into account the difference among the altered counterfactual and the base counterfactual in all the steps.

- **Altered steps difference:** takes into account the difference among the altered counterfactual and the base counterfactual only in the steps affected by the alteration.

These quantities are then divided by the expected difference, which is the product of the alteration value and the number of affected steps. Thus, ideally the final results for both total and altered steps and total differences should be both 1. The final results are obtained by averaging all calculations. For a more formal explanation of these metrics, see appendix A. Even if this metric has been conceived as proxy to evaluate at least one desirable property of counterfactual methods in the case of real world datasets where ground truth is unavailable, it has been also applied to the synthetic and semi-synthetic dataset.

**Metrics from the Axiomatic Definition of Counterfactual.** Recently, the paper Monteiro et al. (2023) has proposed three metrics to measure soundness of a counterfactual inference model without having access to ground truth counterfactuals. Their work is rooted in the Judea Pearl definition of counterfactual (Pearl, 2000), the soundness theorem (Galles & Pearl, 1998), and the completeness theorem (Halpern, 2000), which, together, state that composition, effectiveness and reversibility are necessary and sufficient properties of counterfactuals in any causal model. Let $x$ be an observation with counterfactual parents $\mathbf{pa}$, and $x^*$ a counterfactual of $x$ with parents $\mathbf{pa}^*$. Then, a counterfactual function f can be defined in such a way that $x^* := f(x, \mathbf{pa}, \mathbf{pa}^*)$, where the abduction of the exogenous noise $\epsilon$ is implicit. With this notation, where there is a distinction among the ideal counterfactual function f and its approximation with a counterfactual model $\hat{f}$, Monteiro et al. (2023) define the axioms that an ideal counterfactual function must obey and propose, in relation to each axiom, a metric to evaluate approximated counterfactual functions. In this work, we include these metrics to evaluate counterfactuals in those settings where the ground truth counterfactual is not known but also in the ones where it is known for completeness. The three metrics are the next ones:

**(1) Composition:** Intervening on a variable to have the value it would otherwise have without the intervention will not affect other variables in the system. This implies the existence of a null transformation $f(x, \mathbf{pa}, \mathbf{pa}) = x$ since if $\mathbf{pa}^* = \mathbf{pa}$, then $x$ is not affected. Since the ideal model cannot change an observation under the null transformation, we can measure how much the approximate model deviates from the ideal by calculating the distance between the original observation and the $m$th time null-transformed observation. Given a distance metric $d_x$, such as MAE (which has been selected in this work), an observation $x$ with parents $\mathbf{pa}$ and a functional power $m$ (which is always 1 in this work), we can measure composition as $\mathbf{Composition}^m := d_x\left(x, \hat{f}(x, \mathbf{pa}, \mathbf{pa})\right)$.

**(2) Reversibility:** Reversibility prevents the existence of multiple solutions due to feedback loops. If a mechanism is invertible, this means that if $x^* := f(x, \mathbf{pa}, \mathbf{pa}^*)$, then $x = f(x^*, \mathbf{pa}^*, \mathbf{pa})$. In other words, the mapping between the observation and the counterfactual is deterministic for invertible mechanisms. For a further discussion on this topic, see Monteiro et al. (2023). Thus, we can measure reversibility by calculating the distance between the original observation and the cycled-back transformed observation. Setting $\hat{p}^{(m)}(x, \mathbf{pa}, \mathbf{pa}^*) := \hat{f}\left(\hat{f}(x, \mathbf{pa}, \mathbf{pa}^*), \mathbf{pa}^*, \mathbf{pa}\right)$, given a distance metric $d_x$, an observation $x$ with parents $\mathbf{pa}$ and a functional power $m$ (which is 1 in this work), we can measure reversibility as $\mathbf{Reversibility}^{(m)}(x, \mathbf{pa}, \mathbf{pa}^*) := d_x\left(x, \hat{p}^{(m)}(x, \mathbf{pa}, \mathbf{pa}^*)\right)$. The chosen distance metric in this work is MAE.

**(3) Effectiveness:** Intervening on a variable to have a specific value will cause the variable to take on that value. Thus, suppose Pa is an oracle function that returns the parents of a variable, then we have the following equality: $Pa((f, \mathbf{pa}, \mathbf{pa}^*))$. Effectiveness is difficult to measure objectively without relying on data-driven methods. Following the original paper, we measure effectiveness individually for each parent by creating a pseudo-oracle function $\widehat{Pa}_K$, which returns the value of the parent $pa_K$ given the observation. Using an appropriate distance metric $d_k$, such as accuracy for discrete variables or l1 distance for continuous ones, we measure effectiveness for each parent as $\mathbf{Effectiveness}_k(x, \mathbf{pa}, \mathbf{pa}^*) = d_k\left(\widehat{Pa}_k\left(f_k(x, \hat{pa}_k, pa_k^*)\right), pa_k^*\right)$.

Table 1: Results for time series datasets with both settings $e_f = 0$ and $e_f = 1$ over 10 random seeds. We measure MAE and MBE with respect to the counterfactual ground truth, the total and altered steps differences, and reconstruction and reversibility MAEs and effectiveness accuracy as a fraction. Symbol $\sim$ means that the best results are the more similar ones to the indicated value.

| Metric | Method | Synthetic 0 | Synthetic 1 | Semi-synth. 0 | Semi-synth. 1 | Real world 0 | Real world 1 |
|---|---|---|---|---|---|---|---|
| cf MAE ↓ | LSTM | $.199 \pm .005$ | $.198 \pm .005$ | $.101 \pm .004$ | $.080 \pm .002$ | – | – |
| | CVAE | $.138 \pm .009$ | $.137 \pm .014$ | $.105 \pm .005$ | $.083 \pm .004$ | – | – |
| | CSAE | $\mathbf{.066 \pm .007}$ | $\mathbf{.066 \pm .003}$ | $\mathbf{.070 \pm .003}$ | $\mathbf{.056 \pm .004}$ | – | – |
| cf MBE $\sim 0$ | LSTM | $\mathbf{.001 \pm .011}$ | $\mathbf{.001 \pm .014}$ | $.003 \pm .004$ | $\mathbf{.002 \pm .004}$ | – | – |
| | CVAE | $-.067 \pm .019$ | $.067 \pm .024$ | $.011 \pm .010$ | $-.011 \pm .009$ | – | – |
| | CSAE | $.002 \pm .007$ | $-.002 \pm .015$ | $\mathbf{-.001 \pm .004}$ | $.002 \pm .004$ | – | – |
| Total Steps $\sim 1$ | CVAE | $.457 \pm .091$ | $.443 \pm .066$ | $.037 \pm .011$ | $.045 \pm .110$ | $\mathbf{.899 \pm .024}$ | $1.372 \pm .070$ |
| | CSAE | $\mathbf{.946 \pm .123}$ | $\mathbf{.981 \pm .085}$ | $\mathbf{.747 \pm .052}$ | $\mathbf{.750 \pm .048}$ | $.849 \pm .293$ | $\mathbf{1.183 \pm .098}$ |
| Altered Steps $\sim 1$ | CVAE | $.388 \pm .097$ | $.360 \pm .090$ | $.109 \pm .017$ | $.109 \pm .015$ | $.312 \pm .016$ | $.710 \pm .021$ |
| | CSAE | $\mathbf{.874 \pm .010}$ | $\mathbf{.920 \pm .055}$ | $\mathbf{.300 \pm .015}$ | $\mathbf{.468 \pm .010}$ | $\mathbf{.558 \pm .200}$ | $\mathbf{.794 \pm .111}$ |
| Reconstruction ↓ | CVAE | $.116 \pm .005$ | $.116 \pm .007$ | $.081 \pm.005$ | $.101 \pm .006$ | $.065 \pm .008$ | $.061 \pm .008$ |
| | CSAE | $\mathbf{.051 \pm .006}$ | $\mathbf{.052 \pm .008}$ | $\mathbf{.045 \pm .004}$ | $\mathbf{.059 \pm.004}$ | $\mathbf{.039 \pm .006}$ | $\mathbf{.042 \pm.007}$ |
| Reversibility ↓ | CVAE | $.127 \pm .004$ | $.150 \pm .014$ | $.100 \pm .015$ | $.117 \pm .016$ | $.073 \pm .009$ | $.078 \pm .007$ |
| | CSAE | $\mathbf{.068 \pm .011}$ | $\mathbf{.063 \pm .009}$ | $\mathbf{.050 \pm .004}$ | $\mathbf{.064 \pm .005}$ | $\mathbf{.052 \pm .005}$ | $\mathbf{.054 \pm .004}$ |
| Effectiveness ↑ | CVAE | $\mathbf{1. \pm 0.}$ | $\mathbf{1. \pm .0}$ | $\mathbf{.996 \pm .006}$ | $.991 \pm.004$ | $\mathbf{.631 \pm .005}$ | $\mathbf{.639 \pm .005}$ |
| | CSAE | $.999 \pm.002$ | $1. \pm .0$ | $.992 \pm .007$ | $\mathbf{0.997 \pm .003}$ | $.627 \pm .005$ | $.621 \pm .006$ |

## 4.3 MODELS AND BASELINES

In the time series setting, we compare the counterfactual estimations of CVAE and CSAE for all the metrics described in 4.2, and we add as a benchmark, for the MAE and MBE comparison with ground truth counterfactuals, a LSTM-based conditional forecast model that has as inputs only the historical part of time series $x$ and the value of the event $e$, predicting $y_f$ if $e = e_f$ or $y_{cf}$ if $e = e_{cf}$, where $f$ accounts for factual and $cf$ for counterfactual. Thus, it can be used as a simple time series counterfactual estimator that does not take into account actual values. The encoder and decoder architectures of both CVAE and CSAE are shared, and are based on 1D convolutional and transposed convolutional layers, in a setting inspired in the VAE architecture for time series generation proposed in Desai et al. (2021). To obtain effectiveness metrics, a model based on LSTM layers has been trained to predict the value of the event.

As for the image setting with the color MNIST dataset, we compare CSAE and CVAE for the indicated metrics in 4.2. In this case, the encoder and decoder architectures of both CSAE and CVAE is also shared, and is based on 2D convolutional layers, in a setting based on the VAE architecture of Monteiro et al. (2023). In fact, the CVAE experiment with the color-MNIST dataset is so similar to the normal VAE experiment with the unconfounded dataset in Moteiro et al. (2023), except for the value of the KL hyperparameter. To obtain effectiveness metrics, two models have been trained based on convolutional layers: a classifier to predict the digit and a regressor to predict hue. To train these classifiers, a data augmentation process over MNIST images has been performed. All methods have been implemented with TensorFlow Abadi et al. (2015) and Keras Chollet et al. (2015), using an Adam optimizer with a learning rate of $10^{-4}$ for time series CVAE and CSAE and of $10^{-3}$ for the LSTM and image CVAE and CSAE. Other hyperparameters of CVAE and CSAE such as dimensionality of latent space or the factor of their respective regularizations (KL for CVAE and L1 for CSAE) are particular for each dataset and have been chosen after an optimization process. For more details about implementation, the code of all models is available in the supplementary materials, where it is possible to reproduce all the experiments except the ones involving the real world dataset, as it is protected by the company's privacy policies and safeguarded against unauthorized access.

## 4.4 RESULTS

Table 1 shows the results of time series experiments for the three datasets described in Sec. 4.1.1 in two types of settings: 0 when the factual event $e_f = 0$ and the counterfactual event $e_{cf} = 1$, and 1 when $e_f = 1$ and $e_{cf} = 0$. All values have been obtained after performing 10 experiments with different random seeds, and the intervals correspond to the standard deviation. We see that, in counterfactual MAE metric, CSAE has the best results with an important difference. MBE metric allows to detect biases in the counterfactual estimations. For example, when using CVAE, the inferred counterfactual of time series are often biased towards the actual values, because of its limited

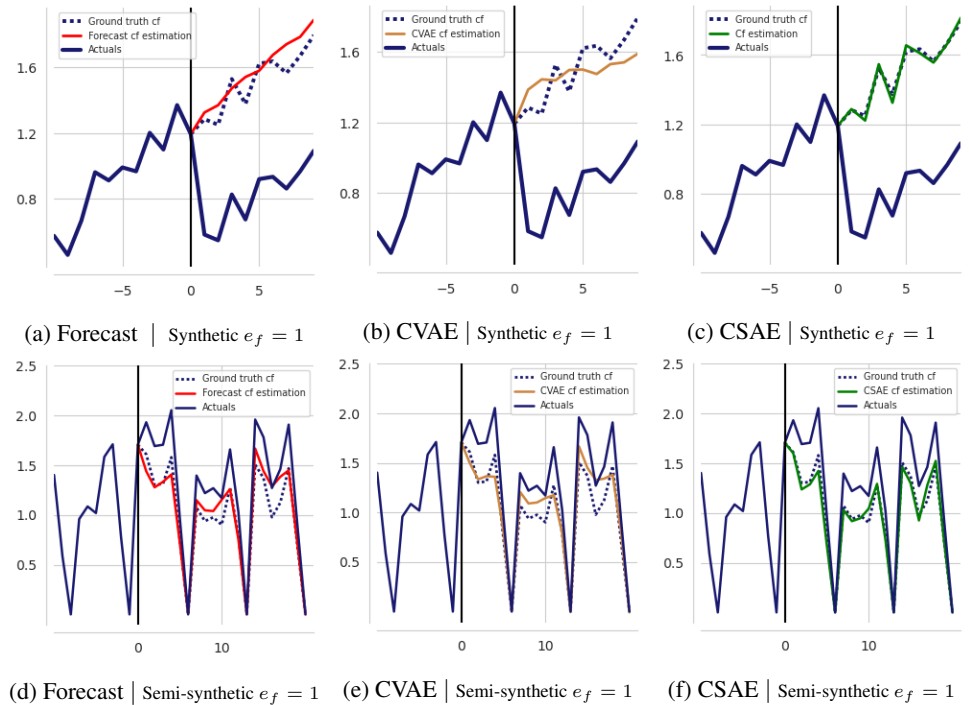

Figure 1: Counterfactual estimation comparison among all the models based on one datum synthetic and semi-synthetic datasets with $e_f = 1$. The vertical line indicates the time step where the event takes place, and the shown historical part is limited to 10 steps.

Table 2: Color MNIST results over 10 random seeds. We measure composition after the null intervention and reversibility after one intervention cycle, effectiveness using digit accuracy, hue absolute error in percentage points (hue$\in [0, 1]$), and MAE with respect to counterfactual ground truth.

| Model | hue intervention ground truth cf. MAE ↓ | null-intervention composition MAE ↓ | digit intervention effectiveness digit Acc.(%) ↑ | digit intervention effectiveness hue MAE ↓ | digit intervention reversibility MAE ↓ | hue intervention effectiveness digit Acc.(%) ↑ | hue intervention effectiveness hue MAE ↓ | hue intervention reversibility MAE ↓ |
|---|---|---|---|---|---|---|---|---|
| CVAE | $4.20 \pm .09$ | $4.15 \pm .09$ | $\mathbf{98.51 \pm .16}$ | $.65 \pm .29$ | $5.32 \pm .08$ | $\mathbf{99.60 \pm .04}$ | $.66 \pm .30$ | $4.86 \pm .10$ |
| CSAE | $\mathbf{2.96 \pm .09}$ | $\mathbf{2.93 \pm .08}$ | $93.01 \pm .93$ | $\mathbf{.51 \pm .21}$ | $\mathbf{4.13 \pm .20}$ | $99.33 \pm .08$ | $\mathbf{.51 \pm .22}$ | $\mathbf{3.59 \pm .13}$ |

disentanglement capacity. This is reflected in MBE metrics, where CSAE and LSTM model have similar values. In composition and reversibility metrics we see that CSAE outperforms CVAE, while in effectiveness metrics results are comparable. Figure 1 shows a comparison among the results of all the methods for one series from the synthetic dataset and one from the semi-synthetic dataset.

In Table 2, that compares CSAE and CVAE in color MNIST, we see that CSAE clearly outperforms CVAE in composition, reversibility and, very importantly, in the ground truth counterfactual. On the other side, CVAE outperforms CSAE in the digit effectiveness with digit intervention. Based on proves with different models and hyperparemeters, there seem to be a trade-off among effectiveness on the one hand and composition, reversibility and ground truth counterfactual on the other side.

## 5 CONCLUSION

In this paper, we have proposed the CSAE, a new autoencoder based model for counterfactual estimation. We have demonstrated that introducing a sparse constraint on an autoencoder results is an effective way to cleanly disentangle the roles of the inputs on the outputs in time series data, which allows to estimate sound counterfactuals. On the other hand, we have shown that this approach is also applicable to image data counterfactuals, where promising results have been obtained. Even if in CSAE, like in CVAE, causal disentanglement is not theoretically guaranteed, the results indicate its utility for counterfactual estimation. Future work could include strengthening the evidence that CSAE is a proper model for image data, expanding the sparsity idea to other deep learning methods or introducing CSAE in more complex SCMs.

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

## A  ADDED VARIATIONS EQUATIONS

Let $y = \{y_t\}, t \in T$ be the time series of the actuals over which we want to perform counterfactuals, $h$ its correspondent historical time series previous to the event, $e_f$ the (factual) event, and $\hat{y}_{cf} = \hat{f}(y, h, e_f, e_{cf})$, where $\hat{f}$ is a counterfactual function and $e_{cf}$ is the counterfactual event, its correspondent counterfactual estimation. Then, we consider a time series $A = 0...0, v_A...v_A, 0...0$ with $T$ steps, where $v_A$ is the value of the alteration which is added only to a certain number of consecutive steps. Let $y^A = y + A$ be the altered time series, then $\hat{y}_{cf}^A$ would be its correspondent counterfactual estimation. We consider that, if our counterfactual model is correct, alterations in the factual time series should be reflected in the counterfactual time series. Thus, ideally $\sum_i \hat{y}_{cf(i)}^A - \hat{y}_{cf(i)} = \sum_i A_i = n_A \cdot v_A$, where $n_A$ is the number of steps affected by the alteration in $A$. Taking into account that we use different time series $A$ with different values $n_A$ and $v_A$, we can express total differences metric for a single time series $y$ (TD) as:

$$TD = \left\langle \frac{\sum_i \hat{y}_{cf(i)}^A - \hat{y}_{cf(i)}}{n_a \cdot v_A} \right\rangle_A, \qquad (3)$$

and altered step differences (ASD) as

$$ASD = \left\langle \frac{\sum_i \hat{y}_{cf(i)}^A - \hat{y}_{cf(i)}}{n_a \cdot v_A} \mathbb{1}_{i \in s_A} \right\rangle_A, \qquad (4)$$

where $s_A$ is the set of altered steps (those with value $v_A$ and not 0) in $A$. We see that, ideally, the result of these averages over the different alteration schemes should be 1. The results given in the paper are the averages of these metrics over all the time series in the test set. The parameters $n_A$, $s_A$ and $v_A$ are particular for every dataset and can be seen in the submitted codes.

## B  COLOR MNIST COUNTERFACTUAL PLOTS

We show several plots of CVAE and CSAE color MNIST counterfactuals and compare them. In Figure 2, we compare composition in CVAE and CSAE with four examples from the color MNIST test dataset. The first images in the left for each subfigure correspond to the same image from color MNIST. Then, in the first row we observe the first 16 null-transformed observations (i.e., 16 iterative reconstructions) with CVAE and, in the second one, the same with CSAE.

In Figure 3, we compare reversibility in CVAE and CSAE with 8 cycled-back transformed observations with random counterfactual parents of hue and digit. Additionally to the cycled-back

observations, we show the counterfactual observations. As in Figure 2, the first images in the left for each subfigure correspond to the same image extracted from color MNIST, the first row corresponds to CVAE transformation and the second one to CSAE.

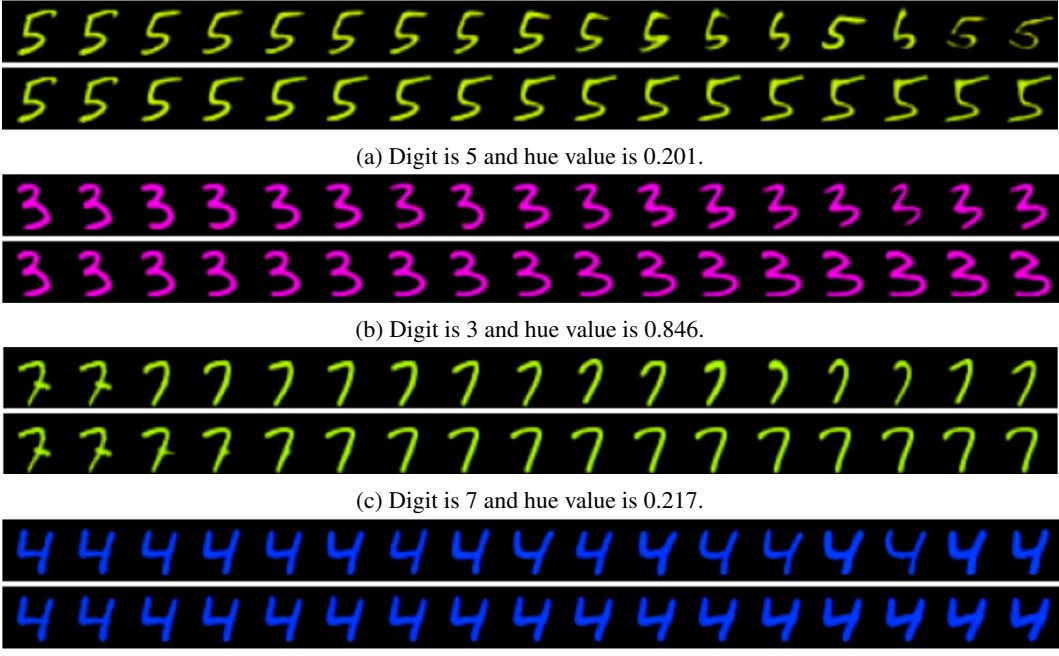

(a) Digit is 5 and hue value is 0.201.

(b) Digit is 3 and hue value is 0.846.

(c) Digit is 7 and hue value is 0.217.

(d) Digit is 4 and hue value is 0.630.

Figure 2: Four examples of 16 null-transformed observations with CVAE (first row in each subfigure) and CSAE (second row). First images in the left are the actual observations.

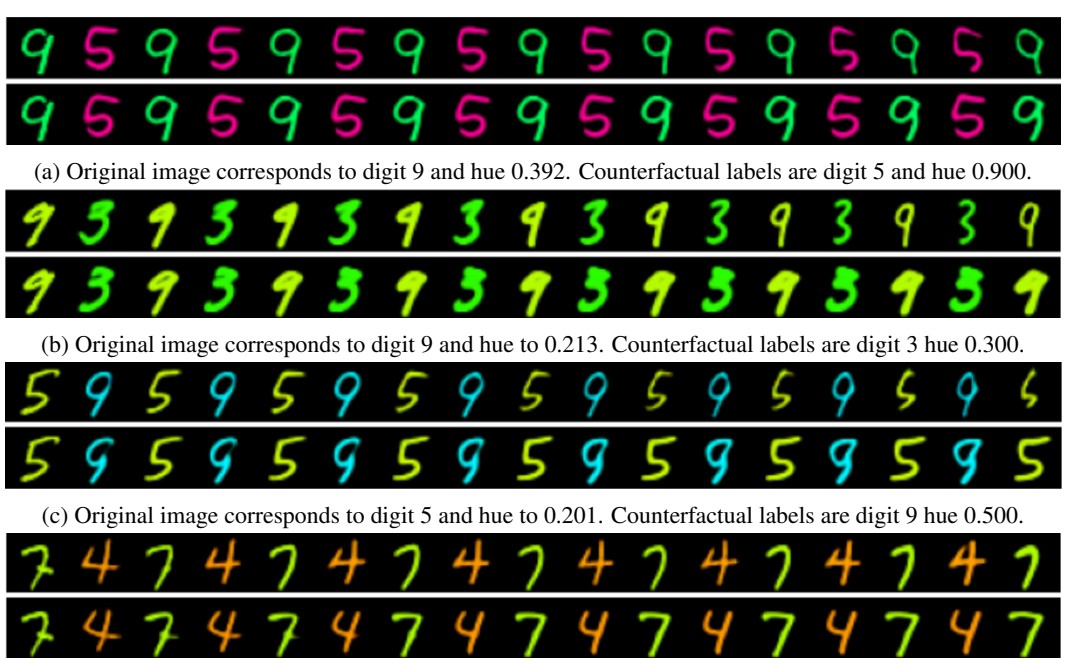

(a) Original image corresponds to digit 9 and hue 0.392. Counterfactual labels are digit 5 and hue 0.900.

(b) Original image corresponds to digit 9 and hue to 0.213. Counterfactual labels are digit 3 hue 0.300.

(c) Original image corresponds to digit 5 and hue to 0.201. Counterfactual labels are digit 9 hue 0.500.

(d) Original image corresponds to digit 7 and hue to 0.217. Counterfactual labels are digit 4 and hue is 0.100.

Figure 3: Four examples of cycled-back transformed observations with random counterfactual parents.First row in each subfigure corresponds to CVAE and second row to CSAE. First images in the left are the actual observations.

