# OpenReview forum: "A Novel Autoencoder Based Approach for Counterfactual Estimation Using Sparsity Constraints"
_ICLR.cc/2024/Conference — ICLR 2024 Conference Withdrawn Submission_

### Official Review · Reviewer_MHr3 · 2023-10-20

**Soundness:** 1 poor
**Presentation:** 2 fair
**Contribution:** 1 poor
**Rating:** 3
**Confidence:** 3

**Summary:**

The paper proposes a conditional sparse autoencoder to perform counterfactual inference on time series and images

**Strengths:**

- The paper is well written in terms of language and organisation
- The method attempts to tackle an important issue , that is counterfactuals in time series

**Weaknesses:**

- The authors have missed a lot of related literature in counterfactuals of timeseries that they should be comparing, contrasting and benchmarking against. For example 3:

    - Continuous-Time Modeling of Counterfactual Outcomes Using Neural Controlled Differential Equations Seedat et al ICML 2022
    - Causal Transformer for Estimating Counterfactual Outcomes, Melnychuk et al ICML 2022
   - Non-parametric identifiability and sensitivity analysis of synthetic control models, Zeitler et al CLeaR 2023
 As a matter of fact the authors completely ignore the entirety of Synthetic Control literature and Epidemiology and Bio-Signals literature that has at its crux the estimation of counterfactual timeseries.

- It is unclear how the proposed method performs the abduction and action step. The only information given is that the representation is sparse. Does this mean that the new counterfactual sample is just dictated by the conditioning factor? Does it include traversing the latent space ?

- It is unclear what kind of causal guarantees the method offers. It appears that the only contribution is a sparsity constraint that is neither novel nor clear how it gives us any causal properties.

- The image task is not properly motivated nor explained. The figure of different color hues is not clear what is the factual part and what is the counterfactual.

- The introduction of sparsity to an autoencoder is not novel as it is well known in the community

**Questions:**

- How does the proposed method compare in theory and practice with synthetic control, and other Neural network and transformer based methods for prediction of time series counterfactuals ?
- How does the method actually guarantee any causal insights ?
- How is the abduction and action performed in this method ?



Overall I dont think this paper is ready yet for publication. Its contributions are not novel, lacking any clear and sound causal guarantees. The evaluation and comparison is insufficient

---

> ### Author Response · Authors · 2023-11-20
> **1/2 Response to reviewer MHr3**
>
> Thanks to the reviewer for taking the time to assess our paper.
>
> Please see below our responses to your comments and questions.
>
> - **Comparison with synthetic control and other methods**: To clarify why we believe that is not reasonable to compare synthetic control models with ours, we add the next sentence in the third paragraph of related work after the sentence “…as it may not always be feasible to access data with a reasonable predictive capacity”:  “For this reason, it is not very reasonable to compare these models with the ones that do not require external information sources, like the one we propose”. In this same paragraph, we cite and briefly discuss [1], which is a clear example of time series counterfactual estimation with synthetic control. As we mention there, the problem with this approach is that it completely relies on the informative capacity of control time series that have not been affected by the event, which could not exist or not be informative enough. Our proposed model, on the contrary, does not require these control time series, and it would not be very useful to compare CSAE against synthetic control models as their capacity to estimate counterfactuals depends more on the data than on the models themselves. Besides this, for our settings, where we do not have synthetic time series, it would be impossible to apply them.
>
>     As for other methods, such as [2] or [3], they are designed to perform counterfactuals at varying time steps, which is not exactly out setting. However, even if abduction-action-prediction scheme is under-explored in time series counterfactuals, future work could include other interesting counterfactual functions based in this scheme such as those based in GANs or diffusion models.
>
> - **Causal guarantees**: We add a clarification of the lack of theoretical guarantees in conclusion after the sentence “…where promising results have been obtained.”:  “Even if in CSAE, like in CVAE, causal disentanglement is not theoretically guaranteed, the results indicate its utility for counterfactual estimation”. It is true that our model does not have theoretical guarantees of causal disentanglement, but many very commonly used models such those presented in [4], [5] or [6], based in generative models such as VAEs, Normalizing Flows or GANs, do not have them either. We offer the intuition of why CSAE disentangles the effects of the inputs in the outputs and why the latent representation tends to not encode the information that the parents bring, and we show empirically that, for our settings, the counterfactual estimations of CSAE are better than those of the benchmark models for most metrics. ”
> - **Abduction and action:** In 3.2, we explain how CVAE performs abduction and action, mentioning that there is a clear parallelism among the abduction-action-prediction scheme explained in 3.1 and the way CVAE preforms counterfactuals. To clarify this with respect to CSAE, we have added the next sentence after the end of the second paragraph of 3.2.1: “As in the case of CSAE, there is a parallelism among the abduction-action-prediction scheme and the way CSAE works to perform counterfactuals; thus, z would make the function of the abducted exogenous noise and the action refers to the introduction of the counterfactual parents **pa*** in the decoder instead of the factual parents **pa**”. We clarify that there is a mistake in the end of this paragraph: the last expression sould be $\hat{x} = D_{\theta}(z, \mathbf{pa^{*}})$ and not $\hat{x} = D_{\theta}(z, \mathbf{pa})$. With that, we hope that it is clear that counterfactual samples are not just dictated by the conditioning factor, but also by the exogenous variable z that encodes the necessary feature of the the input to perform counterfactuals.
> - **Motivation of the image task:** The motivation of the image task is to demonstrate that CSAE is useful as a counterfactual method for these kind of data, apart from time series data. In the penultimate paragraph of introduction, we have changed the last sentence “Additionally, we demonstrate that CSAE is also effective for image counterfactuals” by “Additionally, we demonstrate the general utility of CSAE beyond time series by applying it to an image dataset”.
>
>     As for the images, we have changed the sentence in the second paragraph of the appendix “In Figure 3, we compare reversibility in CVAE and CSAE with 8 cycled-back transformed observations with random counterfactual parents” by “In Figure 3, we compare reversibility in CVAE and CSAE with 8 cycled-back transformed observations with random counterfactual parents of hue and digit”. In the Figure 3 in the appendix we set random counterfactual parents for the hue and the digit, and show counterfactual samples as well as the cycled-back transformations. The labels of subfigures 3a, 3b, 3c and 3d can help see that, but we hope that the change can help to clarify it more.

---

> ### Author Response · Authors · 2023-11-20
> **2/2 Response to reviewer MHr3**
>
> - **Novelty:** In the second paragraph of 3.2.1, we have added this sentence: “Although Sparse autoencoders are well known in the literature, to the best of our knowledge, conditioning them and using the sparse regularization to perform counterfactuals is a novel approach”, with which we hope to have clarified the novelty issue.
>
> [1] Kay H. Brodersen, Fabian Gallusser, Jim Koehler, Nicolas Remy, and Steven L. Scott. Inferring causal impact using bayesian structural time-series models. Annals of Applied Statistics, 9:247–274, 2015.
>
> [2] Continuous-Time Modeling of Counterfactual Outcomes Using Neural Controlled Differential Equations Seedat et al ICML 2022
>
> [3] Causal Transformer for Estimating Counterfactual Outcomes, Melnychuk et al ICML 2022
>
> [4] Nick Pawlowski, Daniel Coelho de Castro, and Ben Glocker. Deep structural causal models for tractable counterfactual inference. Advances in Neural Information Processing Systems, 33:857–869, 2020.
>
> [5] Saloni Dash, Vineeth N Balasubramanian, and Amit Sharma. Evaluating and mitigating bias in image classifiers: A causal perspective using counterfactuals. In Proceedings of the IEEE/CVF Winter Conference on Applications of Computer Vision, pp. 915–924, 2022.
>
> [6] Miguel Monteiro, Fabio De Sousa Ribeiro, Nick Pawlowski, Daniel C. Castro, and Ben Glocker. Measuring axiomatic soundness of counterfactual image models. arXiv preprint arXiv:2303.01274, 2023.

---

### Official Review · Reviewer_cmfR · 2023-10-31

**Soundness:** 2 fair
**Presentation:** 3 good
**Contribution:** 1 poor
**Rating:** 3
**Confidence:** 3

**Summary:**

This paper adds an L1/L2 regularization term over the bottleneck latent variables to the regular Autoencoder loss, Eq. (2).

**Strengths:**

This paper conducts a comprehensive review of Generative Models for counterfactual prediction, and adds an L1/L2 regularization term over the bottleneck latent variables to the regular Autoencoder loss, Eq. (2).

**Weaknesses:**

- The structure of this paper is confusing, leaving me uncertain about the author's intended message and purpose.

- Novelty: The loss function of CSAE is $\mathcal{L}_{\mathrm{CSAE}}(\mathbf{x})=|x-\hat{x}|-\lambda \sum_i\left|z_i\right|$. Is that all? An L1/L2 regularization term? How does it perform counterfactual estimation and what is the counterfactual prediction objective?

- Is Time Series Counterfactual the focus of this manuscript? Why did the authors spend a significant amount of space discussing content that is not directly related to it? It was not until the fourth paragraph that the topic was introduced.

- The motivation of this paper is a bit confusing. What are the challenges in conducting Time Series counterfactuals? How do traditional Time Series methods approach this and what are their limitations? Why can an L1/L2 regularization term over the bottleneck latent variables implement Time Series Counterfactuals? What is the motivation behind this? The presentation should focus more on the core issues addressed in this paper.

- Typos: “… by jointly training and encoder … and a decoder …” → “… by jointly training an encoder … and a decoder …”

**Questions:**

See weakness

**Details Of Ethics Concerns:**

I suspect that this paper was generated solely by AI.

---

> ### Author Response · Authors · 2023-11-20
> **1/2 Response to Reviewer cmfR**
>
> We're grateful to the reviewer for their time spent on our paper.
>
> First of all, we have detected and solved some typos. As for the rest of the comments, please see below our responses to the points pointed out in weaknesses:
>
> - **Novelty:** To clarify how counterfactuals are estimated with CSAE, we have added the next sentence in the end of the second paragraph of 3.2.1: “As in the case of CSAE, there is a parallelism among the abduction-action-prediction scheme and the way CSAE works to perform counterfactuals; thus, z would make the function of the abducted exogenous noise and the action refers to the introduction of the counterfactual parents **pa*** in the decoder instead of the factual parents **pa**”. One of the most used deep learning counterfactual functions for abduction-action-prediction scheme is conditional VAE, which we show that does not perform well enough in time series. To solve that, we propose to use a conditional sparse autoencoder, which, to the best of our knowledge, had not been proposed as counterfactual functions. The way it performs counterfactual estimation is similar to CVAE (see for example [1]), but without the probabilistic nature of the latent variable. It is explained in 3.2.1, but summarizing, the encoder outputs a representation of the input (z), which is then passed to the decoder together with the intervened parent variable, and the final result of this process is the counterfactual estimate. Sparsity regularization enforces the latent space to seize only the necessary information to reconstruct the input, which makes it not capture the information that the parents bring. Thus, we can use a model with the reconstruction capacity of regular autoencoders (usually sparse AEs have similar reconstruction capacity to regular AEs) to perform counterfactuals. Notice that this would be impossible without the sparsity regularization, as the latent space of a normal AE would seize all the information to reconstruct the input and not only the one that is not present in the parents.
>
>     As for the objective, just as in well known counterfactual functions as CVAE, in the training process the model aims to minimize the reconstruction error and, as explained above, the regularization (in the case of CVAE it is the KL with respect to the latent posterior distribution and in CSAE it is L1 or L2 with respect to the latent variable) allows this models to perform counterfactuals. It is impossible to have a direct “counterfactual prediction objective” as, in general cases, we do not know ground truth counterfactuals (the existence our counterfactual ground truth in out synthetic and semi-synthetic settings is just to evaluate the models, as we pretend to simulate real scenarios where those ground truths are not available).
>
> - **Focus of the paper:** To strengthen the idea that the main focus of the paper are time series counterfactuals, we have changed the beginning of the first sentence of third paragraph (in the original version, it was the fourth paragraph), which is now: “Time series counterfactuals, which are the main focus of this paper, can help…”. Apart from this, we have reduced the two first paragraphs of the introduction and united them in a single one.
>
>     Time series counterfactuals is the main focus of the paper, but not the only one as we also have applied our proposed model to image counterfactuals. In first version of the paper, there are two paragraphs for a general introduction to causal deep learning, a third one to present counterfactuals in general and in the fourth we start talking about time series counterfactuals. Now we talk about time series counterfactuals in the third paragraph.

---

> ### Author Response · Authors · 2023-11-20
> **2/2 Response to reviewer cmfR**
>
> - **Motivation and challenges of time series counterfactuals:** In the previous response we explain that the main motivation of this paper are time series and we announce a change in the text to make it clearer. As for the challenges in time series, to avoid misunderstandings we have decided to change the sentence in the second paragraph of page 5 “Even if this is a relevant problem for all kinds of data, it is specially critic for time series since the required precision of the counterfactual is higher than for example in image data” by this new one: “Even if this is a relevant problem for all kinds of data, it is specially critic for some time series settings as the required precision of the counterfactual is specially high”. As explained in the response to reviewer moAT to the question of why we state that the precision required for time series counterfactuals is higher than for other types like image counterfactuals, it stems from a qualitative understanding of what a sound counterfactual estimation is. CVAE, for example, produces reasonably satisfactory counterfactual estimations in most image counterfactual settings. In the case of color-MNIST, for instance, if we want to change the digit, we do not have a highly specific idea of how the counterfactual should be and we will be fulfilled if the counterfactuals resembles the digit that has been selected and some properties like thickness are more or less preserved. For the time series settings that we use in the paper, however, we see that the properties that we want to maintain (i.e. the structure of time series after the intervention) are not specially well preserved with CVAE, at least in comparison with CSAE, as seen in Figure 1. As we say in the responde to reviewer moAT, we recognize that there is some degree of subjectivity in this idea, as for other time series and image settings the precision requirements could change. For this reason, we change the sentence mentioned above.
>
>     One of the most typical time series approach for counterfactual analysis is synthetic control, which is used in [2]. This paper is cited and briefly discussed in the third paragraph of the related work, and we mention that it features an important limitation (which is shared with any other synthetic control approach): it relies on control time series that were predictive with respect to the target before the event and have not been affected by the event. Our paper addresses the problem of time series counterfactual estimation in scenarios were these control time series do not exist or are not enough informative.
>
>     Finally, we hope that the first response helps clarify the function of L1/L2 regularization term.
>
>
> [1]Miguel Monteiro, Fabio De Sousa Ribeiro, Nick Pawlowski, Daniel C. Castro, and Ben Glocker. Measuring axiomatic soundness of counterfactual image models. arXiv preprint arXiv:2303.01274, 2023.
>
> [2] Kay H. Brodersen, Fabian Gallusser, Jim Koehler, Nicolas Remy, and Steven L. Scott. Inferring causal impact using bayesian structural time-series models. Annals of Applied Statistics, 9:247–274, 2015

---

### Official Review · Reviewer_1arB · 2023-11-01

**Soundness:** 1 poor
**Presentation:** 1 poor
**Contribution:** 1 poor
**Rating:** 1
**Confidence:** 4

**Summary:**

This paper presents a conditional auto-encoder with sparsity constraints for counterfactual estimation and generation. The method is motivated for time-series data, but tested on both time-series and image data. Experimental comparisons were made to conditional VAE and, in the setting of time-series data, a LSTM.

**Strengths:**

Counterfactual estimates and generation are relatively under-explored in time-series data. The paper is thus tackling an important and worthy research question.

**Weaknesses:**

This work can be improved in several major areas:

1. While the method is heavily situated within the context of counterfactual estimation and generation, the methodology itself is very marginally tied to causal inference. In fact, it is more related to disentanglement itself. Even for disentangling purpose — the use of sparsity constraint to minimize the information in the bottleneck is heuristic without any theoretical guarantee that z will not attempt to encode information about the conditioning/parent variable (and the success of which should largely depends on the regularization strength). Nothing in this seems to be addressing causal modeling (other than the conditioning), or addresses causal disentanglement at the presence of correlation.

2. In all experimental settings, it is stated that there is no “confounding” in the data and that the two factors (e.g., digit and hue) are independent. This is quite confusing — if two factors have causal relations, there is a high likelihood that they will appear correlated in the data, thus making naive disentanglement (assuming independent generative factors) difficult — In fact, this is the key challenge for most causal inference and generation work to address such “correlation” from observational data. If this correlation does not exist (which I interpret as the confounding as mentioned in the paper), such as estimating intervention effect from randonmized trial data, the the key challenge is gone.

3. Similarly, while the paper was using time-series as a main motivation, pointing out that existing works that deal with images cannot be directly applied to time-series data, it was not pinpointed what exactly are the challenges associated with time-series counterfactuals, and how the presented method addresses them.

4. The work is missing a large number of necessary baselines for comparison, including in time series data (such as RMSN [1], CRN [2], CausalTransformer [3]) and in static image data (causalGAN, causal-VAE, SCM-VAE, ICM-VAE, etc)

[1] Forecasting Treatment Responses Over Time Using Recurrent Marginal Structural Networks
[2] ESTIMATING COUNTERFACTUAL TREATMENT OUTCOMES OVER TIME THROUGH ADVERSARIALLY BALANCED REPRESENTATIONS
[3] Causal Transformer for Estimating Counterfactual Outcomes

**Questions:**

The contribution and rigor of the presented work are overall unclear to me. It’d be helpful if the authors can address my major comments above.

---

> ### Author Response · Authors · 2023-11-20
> **1/3 Response to Reviewer 1arB**
>
> We appreciate the reviewer's effort in reviewing our paper.
>
> Please see below the responses to your comments.
>
> 1. In this work, we present a method which is specifically designed for counterfactual estimation, which is inside causal inference, even if our method can not address other causal inference problems. It is true that there are not theoretical guarantees that CSAE z will not encode information about the parent variables and we just offer the intuition of why this phenomenon is limited with sparse regularization. To clarify the lack of theoretical guarantees of CSAE, we add the following sentence in the conclusion after the sentence “…where promising results have been obtained.”:  “Even if in CSAE, like in CVAE, causal disentanglement is not theoretically guaranteed, the results indicate its utility for counterfactual estimation.” In fact, we do not claim to achieve a perfect disentanglement with our model. However, none of the VAE, Normalizing flows or GAN based model presented, for example, in [1], [2] or [3] has these kind of theoretical guarantees. We consider that the metrics that we use to evaluate counterfactuals estimates are a good indicator of how well a model disentangles information and, more specifically, to which measure the latent variable z of an autoencoder based model like CVAE or CSAE does not encode information about the parent variables. In this sense, we empirically demonstrate that CSAE outperforms CVAE in several datasets, specially in time series settings.
>
>
> 2. To clarify the utility of unconfounded settings to approximate real scenarios, we have added to the TS real world dataset in 4.1.1 the next explanation: “The event here corresponds to the month where a generic treatment enters the market, which usually happens a few months after the date when the patent expires, which is not related with the features of the drug or its sales. Thus, to consider that there is no confounding among the event and other factors that alter sales is a good approximation to reality”. There are several settings where performing counterfactual estimation with variables that do not have confounding can be useful. For example, in CelebA dataset, if we want to perform counterfactuals with glasses or smiling labels, it is a good approximation to consider that there is no confounder neither among those two variables nor among any of them and other important factors such as gender or hair color. In time series, a clear example is the setting of the real world dataset, where the monthly sales of a drug that had been protected by a patent are impacted by the entry to the market of a generic brand of that drug. This usually happens a few months after the patent expires, a phenomenon that is not correlated with any important variable that affects the sales. Thus, to consider that there is no confounding among the event e and factors that affect sales other than e is a good approximation to this real setting.
>
>     On the other hand, we are not proposing a full causal model to work in any desired setting, but a counterfactual function that can be used as a the only component of a counterfactual model for simple settings like the ones used in the paper, or as one of the components of more complex settings. For example, in [1] the authors use image settings where there are causal relations among the different parents of the images and not only among the parents on the one hand and the image on the other hand. To model this, they use normalizing flows for the relations among parents and variational inference (in practice CVAE) for image generation conditioned on parents. In such a setting, CVAE could be substituted by CSAE. In this paper, we have focused in demonstrating the utility of CSAE as a counterfactual function more than in applying it to general causal settings, because we consider that these unconfounded settings demonstrate the utility of CSAE when compared with CVAE or, in case of time series, with an LSTM predictor. Nevertheless, it is true that, in future work, other settings with causal relations among the parents of the final outcome could be also analyzed.

---

> ### Author Response · Authors · 2023-11-20
> **2/3 Response to Reviewer 1arB**
>
> 3. To avoid misunderstandings, we have change the sentence in second paragraph of the introduction (in the original version, it was the third paragraph) “However, these works lack clear mechanisms to disentangle the effects of the causal attributes over the output. For this reason, although the counterfactual images are usually reasonable, it is difficult to transfer these methods to other types of data like time series, where a more disentangled representation of the generating process is required to obtain sound counterfactuals. ” by this new one: “However, these works often do not disentangle enough the effects of the causal attributes over the output and lack precision. For this reason, although counterfactual images are usually reasonable, when applying these methods to some time series settings where more precision is required, the results are usually not satisfactory.” As discussed in **Precision of time series vs other counterfactuals** response to reviewer moAT, for our time series settings we require more precision in counterfactual estimates than for most image settings (even if this does not need to be generally true, as we have acknowledged). On the other hand, we think that most counterfactual methods based on abduction-action-prediction scheme lack the necessary precision for a proper counterfactual estimation in these time series settings. Many of these works are based on VAEs or derivatives of VAEs, and we show that, when directly applying a conditional VAE (which is probably the most common model for abduction-action-prediction in simple causal models) to time series, the results are not satisfactory. We believe that the poor results of the most common counterfactual model when applied to time series is probably the reason why time series is under-explored in counterfactual inference studies (at least, the ones based on abduction-action-prediction scheme), and we propose a model that outperforms CVAE in the analyzed settings. One of the key advantages of CSAE over CVAE is that it eliminates the probabilistic component of  VAEs, responsible for their poorer reconstruction capabilities with respect to AEs, and with the sparsity regularization, it allows a model with the reconstruction capacity of AEs to perform counterfactuals, which, as pointed out in [3], contributes to the quality of the estimated counterfactuals.

---

> ### Author Response · Authors · 2023-11-20
> **3/3 Response to Reviewer 1arB**
>
> 4. As out paper is mainly focused on time series, we have added RMSN [5], CRN [6] and CausalTransformer [7] in the third paragraph of the related work, mentioning that [5] is based on a forecasting method, and [6] and [7] address the problem of treatments that can appear over time with confounded variables. In general, it is difficult to look for benchmarks for the time series settings because this kind of data has been under-explored in counterfactual inference field. For example, one common method for counterfactual estimation is synthetic control  (see e.g. Causal Impact [4]), but it requires a time series which is highly correlated with the one for which we want to measure to effects of the event, whereas our setting is thought for the case where such a time series does not exist or is not informative enough. As for the mentioned papers, for example, RMSN [5] is more about forecasting than counterfactual estimation (we already compare our model with a forecasting model), and CRN [6] and CausalTransformer [7] address the problem of treatments that can appear over time with confounded variables, which is different from our setting, more focused in evaluating a counterfactual function that, as has been shown to work well, could be extended in future work to more complex settings. On the other hand, it is true that, in future work, other counterfactual functions could be included as benchmarks, as those based in normalizing flows or in diffusion models, even if they have not been applied to time series. Apart from that, it would be also interesting to include some of the image counterfactual papers that you mention.
>
> [1] Nick Pawlowski, Daniel Coelho de Castro, and Ben Glocker. Deep structural causal models for tractable counterfactual inference. Advances in Neural Information Processing Systems, 33:857–869, 2020.
>
> [2] Saloni Dash, Vineeth N Balasubramanian, and Amit Sharma. Evaluating and mitigating bias in image classifiers: A causal perspective using counterfactuals. In Proceedings of the IEEE/CVF Winter Conference on Applications of Computer Vision, pp. 915–924, 2022.
>
> [3] Miguel Monteiro, Fabio De Sousa Ribeiro, Nick Pawlowski, Daniel C. Castro, and Ben Glocker. Measuring axiomatic soundness of counterfactual image models. arXiv preprint arXiv:2303.01274, 2023.
>
> [4] Brodersen, K. H., Gallusser, F., Koehler, J., Remy, N., & Scott, S. L. (2014). Inferring causal impact using Bayesian structural time-series models. Annals of Applied Statistics, 9, 247–274
>
> [5] Forecasting Treatment Responses Over Time Using Recurrent Marginal Structural Networks
>
> [6] ESTIMATING COUNTERFACTUAL TREATMENT OUTCOMES OVER TIME THROUGH ADVERSARIALLY BALANCED REPRESENTATIONS
>
> [7] Causal Transformer for Estimating Counterfactual Outcomes

---

### Official Review · Reviewer_moAT · 2023-11-06

**Soundness:** 3 good
**Presentation:** 2 fair
**Contribution:** 1 poor
**Rating:** 3
**Confidence:** 4

**Summary:**

The paper proposes to use conditional sparse autoencoders (CSAE) instead of  conditional variational autoencoders (CVAE) for counterfactual estimation in the DSCM framework focused on timeseries problems. The authors compare the two methods on a synthetic, semi-synthetic and a proprietary timeseries dataset,   as well as coloured MNIST. The experiments indicate superior performance of the CSAE compared to the CSAE.

**Strengths:**

The paper tackles the important problem of counterfactual estimation for time-series problems and identifies performance drawbacks in currently used models.

**Weaknesses:**

The paper is a simple combination of the deep SCM framework with sparse autoencoders. The writing of the paper could use some editing as it's riddled with errors. Furthermore, section 3.1 very closely follows [1] while some of the metrics in 4.2 very closely follow [2], almost being a citation. Even thought the results look promising, the novelty is very limited and the experimental setup is too narrow to provide evidence of this method being suitable for general settings.

[1] Pawlowski, Nick, Daniel Coelho de Castro, and Ben Glocker. "Deep structural causal models for tractable counterfactual inference." Advances in Neural Information Processing Systems 33 (2020): 857-869.
[2] Monteiro, Miguel, et al. "Measuring axiomatic soundness of counterfactual image models." The Eleventh International Conference on Learning Representations. 2022.

**Questions:**

- The paper mentioned that methods are deterministically decoded, and as such does not use well defined probabilities as section 3 suggests. Is this wanted?
- Why is the precision required for timeseries higher than for other counterfactuals?
- The problems mentioned in the paper are already brought up in [2] (see the confounded data experiments) and have been tackled in e.g. [3]. How does this method compare?
- Why does the probabilistic nature of CVAEs introduce additional errors?
- The explanations of the metrics are hard to follow. It would be helpful to add equations here, especially for the "Added variations"
- Whats the assumed causal graph for the experiments?

[3] Kumar, Amar, et al. "Debiasing Counterfactuals in the Presence of Spurious Correlations." Workshop on Clinical Image-Based Procedures. Cham: Springer Nature Switzerland, 2023.

---

> ### Author Response · Authors · 2023-11-20
> **1/2 Response to the Reviewer moAT**
>
> We thank the reviewer for their time in reading and reviewing our paper.
>
> We are working on improving the style of the text and to solve some the errors that you mention. In our work, we present a novel counterfactual function based on sparse autoencoders that leverages the reconstruction capabilities of AEs (in contrast to the poorer reconstruction capabilities of VAEs) to perform counterfactuals and show good results, specially on time series. Section 3.1 presents a background on structural causal models closely following [1], which is cited, as it offers a proper introduction for the reader to SCMs. On the other hand, we mention that we extract some of the metrics [2] as they are useful to evaluate counterfactuals.
>
> Please see below our responses to your questions.
>
> - **Deterministically decoding**: The fact that CSAE is not deterministic is wanted. In the point **Errors in CVAE due to its probabilistic nature** we explain better why the probabilistic nature of VAEs make its reconstruction worst, and mention a change in the text to clarify that. One important difference among our proposed method CSAE and CVAE is the fact that CSAE works as an AE with a sparse regularization whereas CVAE is a conditioned VAE. In general, reconstruction capabilities of regular/sparse AEs are much higher than the ones of VAEs, which is caused mainly by the probabilistic nature of VAEs, as it is easier for the model to properly reconstruct data if every datum is mapped to a point in the latent space than if it is mapped to a distribution from which a random point will be selected to be the input of the decoder. On the other hand, for a counterfactual model, generally the reconstruction capability is closely related to how well it performs counterfactuals. Furthermore, we empirically show that sparse regularization allows a model with the reconstruction prowess of AEs (in constrast to VAEs) to perform counterfactuals, which improves the metrics when compared with VAE based frameworks.
> - **Precision of time series vs other counterfactuals**: We have changed the statement in the third paragraph of 3.2.1: “Even if this is a relevant problem for all kinds of data, it is specially critic for time series since the required precision of the counterfactual is higher than for example in image data.” by this other one: “Even if this is a relevant problem for all kinds of data, it is specially critic for some time series settings as the required precision of the counterfactual is specially high”. The original statement stems from a qualitative understanding of what a sound counterfactual estimation and we have decided to change it as we recognize that it has some degree of subjectivity. In spite of this correction, next we explain the reasons why the statement was made. CVAE produces reasonably fulfilling counterfactual estimations in most image counterfactual settings. In the case of color-MNIST, for instance, if we want to change the digit, we do not have a highly specific idea of how the counterfactual should be and we will be fulfilled if the counterfactuals resembles the digit that has been selected and some properties like thickness are more or less preserved. For the time series settings that we use in the paper, however, we see that the properties that we want to maintain (i.e. the structure of time series after the intervention) are not specially well preserved with CVAE, at least in comparison with CSAE, as seen in Figure 1. However, for other time series and image settings the precision requirements could change.

---

> ### Author Response · Authors · 2023-11-20
> **2/2 Response to the Reviewer moAT**
>
> - **Comparison with other works**: the main contribution of our work is in time series. To clarify that, we have changed the beginning of the first sentence of fourth paragraph, which is now: “Time series counterfactuals, which are the main focus of this paper, can help…”. Despite being time series the main focus of our work, however, it is true that, as we also perform image counterfactuals with CSAE to demonstrate its applicability to this kind of data, some comparisons can be made with some papers that apply counterfactual models to images.
>
>     The color-MNIST experiment is very similar to the unconfounded setting of the color-MNIST dataset in [2]. We have added the sentence: “In fact, the CVAE experiment with the color-MNIST dataset is so similar to the normal VAE experiment with the unconfounded dataset in Moteiro et al. (2023), except for the value of the KL hyperparameter.” in the second paragraph of 4.3 to show that CVAE is almost a reproduction of one of the models that they use. We demonstrate that CSAE outperforms our CVAE version and, seemingly, also outperforms the GAN based model that they use in most metrics.
>
>     As for [3], we have added in the second paragraph of related work the next sentence: **“**Kumar et. al (2023) use a GAN based approach to address the specific problem of spurious correlations in medical datasets**” .** The problem with this paper is that it performs very different experiments, datasets have little to do with ours, and they address the specific problem of spurious correlations, which makes it very difficult to compare, even if in future work it could be interesting to apply cycle GAN, the method that they use, to color-MNIST and compare it with CSAE and CVAE.
>
> - **Errors in CVAE due to its probabilistic nature**: with respect to this, we have added an explanation in the penultimate paragraph of 3.2.3: “the probabilistic nature of CVAE introduces additional errors as the input of the decoder comes from a random sample of the posterior distribution that the encoder outputs and is not directly the outcome of the encoder.”. This question is very related with the first point: basically, VAEs tend to have worst reconstruction capabilities than AEs due to the fact that the input of the decoder is a random point of the distribution that the encoder outputs and not directly a deterministic point in latent space that stems from the encoder. On the other hand, as explained in our paper (3.2) and in [2], the reconstruction capacity of a counterfactual model is closely related to the quality of its counterfactual estimates.
> - **Metrics equations**: we have includded a formal explanation of the added variation metrics in an appendix. Whit this, we hope to clarify these metrics.
> - **Causal graph of the experiments**: to help better understand this, we have changed, both in 4.1.1 and 4.1.2, the abbreviation SCM by “causal graph”, which is what we are really describing, since SCM include the structural equations apart from the causal graph. As mentioned in these sections, in the case of time series datasets, the time series y has two independent parents which are h (the historical time series previous to the event) and e (the event). For the color-MNIST dataset, the image has two independent parents which are the digit and the hue.
>
> [1]Nick Pawlowski, Daniel Coelho de Castro, and Ben Glocker. Deep structural causal models for tractable counterfactual inference. Advances in Neural Information Processing Systems, 33:857–869, 2020.
>
> [2] Monteiro, Miguel, et al. "Measuring axiomatic soundness of counterfactual image models." The Eleventh International Conference on Learning Representations. 2022.
>
> [3] Kumar, Amar, et al. "Debiasing Counterfactuals in the Presence of Spurious Correlations." Workshop on Clinical Image-Based Procedures. Cham: Springer Nature Switzerland, 2023.